# Experimental Research on Shear Failure Monitoring of Composite Rocks Using Piezoelectric Active Sensing Approach

**DOI:** 10.3390/s20051376

**Published:** 2020-03-03

**Authors:** Yang Liu, Yicheng Ye, Qihu Wang, Weiqi Wang

**Affiliations:** 1Resources and Environmental Engineering Institute, Wuhan University of Science and Technology, Wuhan 430081, China; liuyang@wust.edu.cn (Y.L.); yeyicheng@wust.edu.cn (Y.Y.); wangweiqi@wust.edu.cn (W.W.); 2Hubei Key Laboratory of Mechanical Transmission and Manufacturing Engineering, Wuhan University of Science and Technology, Wuhan 430081, China; 3Industrial Safety Engineering Technology Research Center of Hubei Province, Wuhan 430081, China

**Keywords:** piezoelectric active sensing, composite rocks, shear failure, damage detection, damage index

## Abstract

Underground space engineering structures are generally subject to extensive damages and significant deformation. Given that composite rocks are prone to shear failure, which cannot be accurately monitored, the piezoelectric active sensing method and wavelet packet analysis method were employed to conduct a shear failure monitoring test on composite rocks in this study. For the experiment, specimens were prepared for the simulation of the composite rocks using cement. Two pairs of piezoelectric smart aggregates (SAs) were embedded in the composite specimens. When the specimens were tested using the direct shear apparatus, an active sensing-based monitoring test was conducted using the embedded SAs. Moreover, a wavelet packet analysis was conducted to compute the energy of the monitoring signal; thus allowing for the determination of the shear damage index of the composite specimens and the quantitative characterization of the shear failure process. The results indicated that upon the shear failure of the composite specimens, the amplitudes and peak values of the monitoring signals decreased significantly, and the shear failure and damage indices of the composite specimens increased abruptly and approached a value of 1. The feasibility and reliability of the piezoelectric active sensing method, with respect to the monitoring of the shear failure of composite rocks, was therefore experimentally demonstrated in this study.

## 1. Introduction

There have been significant recent developments in underground space engineering. Accordingly, the highly complex conditions of geological bodies have been reported, especially at the interfaces where different rock masses are in contact and are widely distributed in the crust of the Earth. In general, two or more rock masses are mutually embedded, which is referred to as a composite rocks formation [1,2]. There are significant differences between the physical and mechanical properties of different rock masses. When a tunnel or a roadway intersects a contact zone, the rock masses in contact undergo uncoordinated deformation, which generates shear stress, thus resulting in the shear failure of the rock masses [3,4,5]. Moreover, the findings of previous researches revealed that the roadway of a contact zone is subject to extensive shear failure due to uncoordinated deformation, which is a major safety hazard with respect to the normal operation of underground space engineering structures. The existing monitoring methods for the measurement of the macroscopic displacement changes cannot be employed to determine the shear failure process of composite rocks masses, characterize the extent of the damage, and issue timely warnings [6,7]. It is therefore necessary to conduct further research on the shear failure monitoring of composite rocks.

At present, studies have been conducted with respect to the stability evaluation of rocks [8,9] and rock failure monitoring [10,11,12]. With respect to rock failure monitoring, there are four main methods: (1) the monitoring of micro-seismic signals to conduct a preliminary analysis of the rock failure characteristics [13,14]; (2) the monitoring of acoustic emission signals to investigate the characteristics and surface degradation of soft rock, in addition to the development of a roughness damage model based on the acoustic emission time characteristics [15]; (3) the use of acoustic waves for the evaluation of the rock mass characteristics in the damaged area of the rock slope excavation [16]; and (4) the use of fiber Bragg grating (FBG) sensors for the monitoring of the surrounding rock stability of tunnels [17]. The above methods allow for the accurate monitoring of rock mass failure in various applications; however, dynamic monitoring methods with respect to the rock mass failure process require further investigation, especially the monitoring of the interfacial damage of composite rocks masses.

With similar properties to those of rocks, concrete materials, which are widely used in civil infrastructure as substitutes, are subject to significant deterioration due to adverse service conditions such as corrosion [18,19,20] and vibration [21,22]. To mitigate the deterioration of infrastructure, there have been significant developments with respect to structural health monitoring (SHM) [23,24,25,26,27,28] in recent decades, and piezoceramic materials have been widely applied for the SHM of concrete structures [29,30,31] due to their low cost, high sensitivity [32], rapid response [33,34], wide frequency range [35,36], and energy harvesting capacity [37,38,39], in addition to actuating and sensing capacities [40,41,42]. Song et al. developed a smart aggregate (SA), which was fabricated by the embedment of waterproof piezoceramic patches in small concrete or marble blocks [43,44], for the SHM of concrete structures. Smart aggregates can significantly increase the applicability of piezoceramic transducers to concrete structures [45,46,47,48]. Moreover, SAs can transmit and detect stress wave signals, and they have received extensive research attention for concrete structure applications with respect to early age hydration monitoring, impact detection, and crack monitoring [49,50]. Piezoceramic transducers can realize active sensing and electromechanical impedance (EMI) monitoring based on SHM [51,52,53,54]. Wu et al. successfully applied the EMI method and wave analysis method to the monitoring of soft interlayer landslides, and obtained reliable experimental results [55,56]. Based on the wavelet packet analysis method, Xu et al. conducted a study on piezoelectric active sensing for the detection of the interfacial de-bonding of concrete-filled steel tubes, and obtained reliable experimental results [57]. In general, piezoelectric active sensing is suitable for the monitoring of interfacial damage and bond slips [31,45,58].

In this study, the piezoelectric active sensing and wavelet packet analysis methods were used to conduct the interfacial shear failure monitoring of composite rock specimens, which were fabricated using two types of concrete materials with different strengths. In the fabricated composite specimens, two pairs of SAs were mounted symmetrically on both sides of the interface. An SA in each pair was used as a driver for the transmission of a frequency sweep signal through the interfaces between the different rocks. During the shear test, upon interfacial damage, signal attenuation occurred. The other SA was used as a sensor for the reception of signals. Wavelet packet analysis was then conducted for the computation of the received signal energy, and the failure process of the composite rocks were analyzed in accordance with a decrease in energy.

This manuscript is organized as follows. Section 1 introduces the research background. Section 2 presents the monitoring principle of the piezoceramic active sensing method, followed by a description of the experimental setup and procedures in Section 3. In Section 4, an analysis of the test results is presented; followed by the conclusions in Section 5, in addition to the scope of future research.

## 2. Aims of Research and Principle of Detection

### 2.1. Aims of Research

There are three aims of this study. Firstly, in view of the lack of monitoring methods for composite rocks failure, the feasibility of piezoelectric active sensing method applied to composite rocks failure monitoring is studied. Secondly, the damage index of composite rocks are quantitatively characterized by wavelet packet energy method. Finally, the feasibility and reliability of the method are further verified by repeated tests.

### 2.2. The Piezoceramic Transducers

As a unique type of piezoelectric ceramic material, lead zirconate titanate (PZT) is widely employed in the SMH field due to its significant piezoelectric effect. The fragility of PZT materials is usually encapsulated without changing its function in engineering testing to meet different operating conditions. Song et al. proposed an SA that was packaged by embedding PZT patches into marble or concrete blocks; and can be further embedded into a concrete structure for multiple-purpose monitoring [43,44,58]. Given that the piezoceramic patch demonstrates direct and inverse piezoelectric effects, the packaged SA can realize dual functions of the transmission and reception of stress wave signals. The SA used in this study was a multifunctional sensor based on piezoceramics, which is shown in Figure 1. The SA was packaged by sandwiching two PZT patches between a pair of cylindrical marble blocks with epoxy resin. Moreover, it had a diameter of 25 mm and height of 20 mm, as shown in Figure 2. The diameter and thickness of the PZT disk are 15 mm, and 0.3 mm, respectively, while the elastic modulus and density are 56 Gpa and 7500 kg/m3, respectively. A BNC connector with a cable provides the electric connection to the smart aggregate.

### 2.3. Principle of Piezoelectric Active Sensing Method

In this study, an active sensing method based on piezoelectric SAs was employed for the monitoring of the shear failure of composite rocks. Figure 3 presents the schematic for the monitoring of the shear failure of the composite rock based on the active sensing method. In this method, two pairs of piezoelectric SAs are installed on both sides of the interface of the composite rocks. One SA is used as an actuator under a swept sine wave excitation, and the induced stress wave is propagated through the interface. Moreover, the other SA serves as a sensor for the detection of the stress wave signals. Upon the occurrence of shear failure, the propagation medium of the wave is discontinuous due to the unevenness of the interface. Thereafter, the amplitude, peak value, and energy of the received signal are attenuated. The energy of the monitoring signal is then computed according to the wavelet packet analysis method, and the shear failure of the composite rocks are determined in accordance with a decrease in energy. The amplitude, peak value, and energy of the monitoring signal, among other characteristics, are analyzed to monitor the failure of the composite rocks.

### 2.4. The Shear Failure Damage Index of Rock

The wavelet packet analysis method, as a signal processing tool, was employed in this study to process and analyze the detected signal. The wavelet packet analysis facilitates the decomposition of the signal into different frequency bands with respect to the any time-frequency resolution, which has the characteristics of accurate subdivision and strong time-frequency localization [57]. Based on the wavelet packet analysis, an equation was established for the computation of the shear failure damage index of rock, to quantitatively represent the relative loss value of the transmitted signal energy during the shear failure development of the rock.

Under the assumption that the signal detected by the sensor is denoted by *V*; the *n*-level wavelet decomposition is used to decompose the signal into a *2^n^* signal, which can be denoted as $v_{1},v_{2},\ldots ,v_{j},\ldots ,v_{2}^{n}$. Moreover, $v_{j}$ represents the monitoring signals after decomposition, which can be expressed as follows:(1)$$v_{j}=\left[v_{j,1},v_{j,2},\ldots ,v_{j,m}\right]$$

In this equation, *m* is the number of the specimen data-points, and *j* is the number of frequency bands ($j=1,2,\ldots ,2^{n}$).

The energy of each decomposition signal $M_{j}$ is defined as:(2)$$M_{j}=v_{j,1^{2}}+v_{j,2^{2}}^{}+\cdots +v_{j,m^{2}}$$

The energy vector of the signal $v_{i}$ during the *i*-th monitoring is obtained by the following equation:(3)$$M_{i,i}=\left[M_{i,1},M_{i,2},\ldots ,M_{i,2^{n}}\right]$$

The root mean square deviation (RMSD) is used to compute the shear damage index of rock, i.e., the degree of rock shear damage at different instances in time. The indicator is defined by computing the RMSD between the initial state during the test and the energy vector of the signal detected during the subsequent state during the test. Based on the wavelet packet analysis method, the shear failure damage index of rock is computed at the *i*-th instance as follows:(4)$$Y=\sqrt{\frac{\sum {}_{j=1}^{2^{n}}{\left(M_{i,j}-M_{h,j}\right)}^{2}}{\sum {}_{j=1}^{2^{n}}{\left(M_{h,j}\right)}^{2}}}$$

In the equation, $M_{h,j}$ is the energy value of the *j*-th frequency band in the initial state or the healthy state ($j=1,2,\ldots ,2^{n}$), and $M_{i,j}$ is energy value of the monitoring signal at the *i*-th instance.

The shear damage index of rock can be used to characterize the degree of rock shear failure development and determine whether the rock is completely damaged based on the energy loss trend, thus providing a timely warning.

## 3. Experimental Setup and Methodology

### 3.1. Specimen Preparation and SA Installation

Given the difficulty of obtaining composite specimens in the contact zone, in addition to the good repeatability and mechanical stability of physically similar specimens; P325 Portland cement, river sand, gypsum powder with particle sizes of 0.9–1.2 mm, and water were selected to produce the similar composite specimens [59,60]. Given that the proportions of the materials were different, the physical and mechanical parameters of the specimens were different; thus, different proportions of cement mortar can be used to produce different composite specimens. The dimensions of the designed composite rock specimens were 100 mm × 100 mm × 100 mm, as shown in Figure 4. Composite specimens were poured into the upper and lower layers with two different proportions of single materials. The proportions of the two materials accounted for 50% of the volume, and the contact angle was 0°. The proportions of the two materials and corresponding mechanical parameters are shown in Table 1. Three composite specimens were poured into a mold with dimensions of 100 mm × 100 mm × 100 mm. The mold was fixed on the vibration table, and the specimens were cured for 28 days, as shown in Figure 5.

Based on the schematic diagram and drawings (see Figure 4), the three specimens were processed with two pairs of symmetrical holes for each specimen. The diameters and depths of the holes were 26 mm and 22 mm, respectively, as shown in Figure 6. Thereafter, SAs were installed symmetrically in the holes of the three specimens, and cement was used as binder for curing for 7 days, as shown in Figure 7.

### 3.2. Test Device and Parameter Setting

The test device consisted of a direct shear apparatus, a control system, a direct shear box, piezoelectric monitoring equipment, a control system for the monitoring equipment, and composite specimens with SAs, as shown in Figure 8. We used the SC-HY-PZT-2.0 multifunction piezoelectric signal monitoring equipment, manufactured by Jiang Susan Chuan Intelligent Technology Co., Ltd., China, that includes a host and piezoelectric signal measurement and control software. The control software controls the host for signal excitation and acquisition. It should be noted that the direct shear apparatus control system and monitoring equipment control system can collect corresponding data independently. During the test, the piezoelectric monitoring equipment emitted a sweep sine wave through the transmitting channel. The frequency range of the sweep sine wave was 0.1–150 kHz. The frequency increment was 10 KHZ, and the step time was 0.2 ms. The amplitude and period of the signal were 10 V and 3.2 ms, respectively. The sampling frequency, sampling length, and sampling time of the data acquisition system were 2 MHZ, 16 K, and 8.192 ms, respectively. During the test, Actuators SA 1 and SA 2 transmit their signals to Sensors SA 3, and SA 4, respectively. No interference was observed between the two signals.

### 3.3. Test Steps

As shown in Figure 8, a YAW-1000A microcomputer-controlled electro-hydraulic servo shear apparatus was used to conduct the shear test on the composite specimens. The normal load was 25 KN, and the specimens were sheared at a rate of 0.002 mm/s. In intervals of 2 min, each driver successively generated sweep sine waves. Moreover, the corresponding sensor monitored the propagation of waves through the interface of the composite specimen within the acquisition system until the completion of each shear test.

## 4. Experimental Results and Discussions

### 4.1. Analysis of the Shear Test Results of Composite Specimens

The shear stress-displacement curve and shear stress-time curve of Composite specimens I, II, and III are shown in Figure 9, Figure 10 and Figure 11, respectively. The shear failure process of the composite specimen was divided into two stages. In Stage 1, the shear stress increased linearly and reached the peak value. In Stage 2, the shear stress of the curve abruptly decreased, and the interface of the composite specimen underwent shear failure. The shear failure of Composite specimen I occurred after 22 min, the shear failure of Composite specimen II occurred after 19.8 min, and the shear failure of Composite specimen III occurred after 20.4 min. Although all specimens are made from similar materials, the failure time of interfaces will be slightly different due to test error and individual difference. Figure 9, Figure 10 and Figure 11 reveal that the three composite specimens underwent shear failure after approximately 20 min at shear displacements of approximately 2.5 mm; thus, the results for the three composite specimens were in good agreement. The composite specimens after shear failure are shown in Figure 12. As can be seen from the figure, the three composite specimens exhibited uneven damages along the interface. Generally, the composite rocks will be damaged along or near the interface. Due to the bonding degree of the interface and individual differences, the specimen will be damaged unevenly along the interface. Although the composite specimen materials, pouring methods and curing time are the same, the failure interface of the specimen will be slightly different and they are all near the interface.

### 4.2. Time-Domain Analysis of Monitoring Signals

During the shear tests, each actuator successively generated sweep sine wave signals at intervals of 2 min, and the corresponding sensors monitored the waves transmitted through the interface of the composite specimens. In the study, 0.001–0.004 s signals with obvious monitoring signal characteristics were selected for analysis. The peak curve of the amplitude of each received SA signal in Composite specimen I is shown in Figure 13. The peak value of the amplitude of the monitored signal decreased abruptly after 22 min, which was consistent with the shear failure that occurred in Composite specimen I after 22 min. The peak curve of the amplitude of each received SA signal in Composite specimen II is shown in Figure 14. The peak value of the amplitude of the monitoring signal decreased abruptly after 20 min, which was in good agreement with the shear failure of Composite specimen II after 19.8 min. Due to the uneven pouring of Composite specimen II and the different density of SAs’ position, the amplitude of signals received by the two sensors is quite different, but the change trend is similar. The peak curve of the amplitude of each received SA signal in Composite specimen III is shown in Figure 15. The peak value of the amplitude of the monitoring signal decreased abruptly after 22 min, which was in good agreement with the shear failure of Composite specimen III after 20.4 min. Based on the analysis of the test results, the peak value of the amplitude of each monitoring signal was found to correspond to the shear failure process of the composite specimen. During the shearing process, the peak values of the amplitudes of the monitoring signals gradually decreased. When the specimens underwent complete shear failure, the peak values of the signal amplitudes decreased abruptly.

Based on the results of the signal analyses before and after the shear failure of the composite specimens, the signals received by the SAs before and after the shear failure of the Composite specimen I are shown in Figure 16. The amplitude of the signal detected by the two sensors after 22 min was significantly lower than that of the signal detected by the two sensors after 20 min. The signals received by the SAs before and after the shear failure of Composite specimen II are shown in Figure 17. The amplitude of the signal detected by the two sensors after 20 min was significantly lower than the amplitude of the signal detected by the two sensors after 18 min. The signals received by the SAs before and after the shear failure of Composite specimen III are shown in Figure 18. The amplitude of the signal detected by the two sensors after 22 min was significantly lower than that of the signal detected by the two sensors after 20 min. The analysis reveals that after the composite specimens underwent shear failure, the amplitude of the interface wave significantly decreased. The resulting time-domain signal reveals that this method can be used to monitor the shear failure of composite rocks. Compared with the traditional strain monitoring method, when the composite sample is damaged, the stress waves value drops obviously, which can help to send out the warning signal, and also shows that the method has high sensitivity.

### 4.3. Rock Shear Damage Index Based on Wavelet Packet

For the quantitative analysis of the entire rock shear failure process, the energy of each monitoring signal was computed based on the wavelet packet energy analysis method. Moreover, the rock shear damage index was computed based on Equations (1)–(4). The shear failure damage index of Composite specimen 1 monitored by SA 3 is shown in Figure 19a, and the shear failure damage index of Composite specimen I monitored by SA 4 is shown in Figure 19b. The figures demonstrate that during the entire shearing process, the shear fracture damage index of Composite specimen I monitored by both sensors gradually changed, and then increased abruptly when Composite specimen I underwent failure after 22 min. The shear failure damage indices of Composite specimen II monitored by SA 3 and SA 4 are shown in Figure 19c,d, respectively. There was an initial gradual increase in the shear failure damage index of Composite specimen II, followed by an abrupt increase after the shear failure that occurred after 19.8 min. The shear failure damage indices of Composite specimen III monitored by SA 3 and SA 4 are shown in Figure 19e,f, respectively. There was an initial gradual increase in the shear failure damage index of Composite specimen III, followed by an abrupt increase after the shear failure that occurred after 20.4 min. The variation of the shear failure damage index of the three composite specimens exhibited high repeatability and consistency. Therefore, the shear failure damage index of the composite specimens based on the wavelet packet analysis can quantitatively characterize the shear failure process of the composite specimens; effectively monitor the shear failure of the composite specimens; and provide a timely warning when the damage index abruptly increases and approaches a value 1. If this method can be applied to the actual engineering monitoring, it can ensure the safe operation of the underground space structure engineering. However, due to the limitation of practical engineering conditions, the research of the application of this method needs to be further conducted.

The general monitoring method cannot quantitatively characterize the damage index. The study uses the wavelet packet energy method to calculate the stress wave energy value, and uses the root mean square difference between the energy to characterize the damage index. The experimental results of this research and the results of applying the method to the degree of debonding of the reinforced concrete interface and the degree of landslide slip are consistent, which again proves the feasibility of this method [55,57].

### 4.4. Further Discussions

When the composite specimens fail, the shear stress drops rapidly, as shown in Figure 9, Figure 10 and Figure 11. This indicates that the composite specimens have brittle failure. This also causes the stress wave to drop rapidly. By analyzing the peak value of stress wave, the damage process of composite specimens can be described to some extent, but sometimes the peak value will fluctuate, which will affect the accuracy of the test results. From Figure 12, Figure 13 and Figure 14, it can be seen that the time-domain signal of composite specimens after failure declines significantly, so the failure process of composite specimens can be monitored from the time-domain signal analysis. However, the failure damage index of composite specimens cannot be quantitatively characterized. Wavelet packet energy method is used to calculate the energy value of monitoring signal, and root mean square deviation is used to calculate the shear damage index of rocks, which can quantitatively represent the failure degree of composite samples. When the composite specimen is damaged, the damage index is shown in Table 2. In conclusion, the monitoring of composite rocks shear failure based on piezoelectric active sensing method has potential engineering application value.

## 5. Conclusions and Future Work

The piezoelectric active sensing monitoring method and wavelet packet energy analysis method were employed for the shear failure monitoring of rock composites fabricated in this study. The specimens were drilled and embedded with piezoelectric SAs. A shear test was carried out using direct shear apparatus. Meanwhile, the monitoring test was carried out by transmitting sweep sine wave and receiving wave. The shear stress-time curves of the composite specimens, in addition to the amplitude peak curves of the signals detected by the SAs before and after the shear failure of the composite specimens, were compared and analyzed. Upon the shear failure of the composite specimens, there were corresponding changes in the amplitude and peak value of the signal. This consistency proves the feasibility of the piezoelectric active sensing method in the shear failure monitoring of composite rocks in the laboratory condition. However, the actual engineering conditions are complicated, and the research of the application of this method needs to be further carried out. The damage index of the rock shear failure was computed using the wavelet packet analysis method to quantitatively represent the process of rock shear failure. The variation trends of the damage index and shear stress with respect to time were consistent. In addition, the good repeatability of the test results of the three composite specimens further indicates the feasibility and reliability of the method. In future work, the stress wave propagated across the interface in the composite rocks will be modeled based on the fractal contact theory, which was recently developed for the modeling of stress waves propagated through the interfaces of bolted joints [61,62].

## Figures and Tables

**Figure 1 sensors-20-01376-f001:**
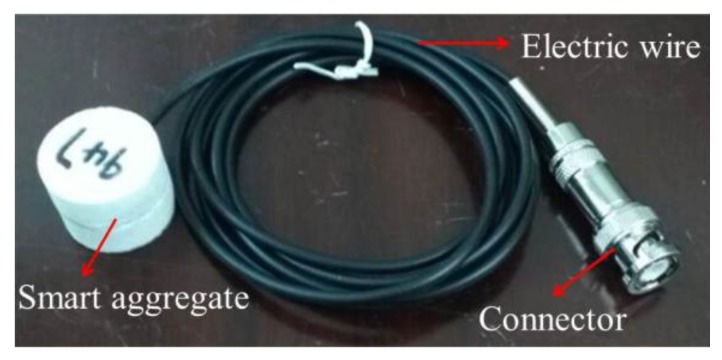
A smart aggregate (SA).

**Figure 2 sensors-20-01376-f002:**
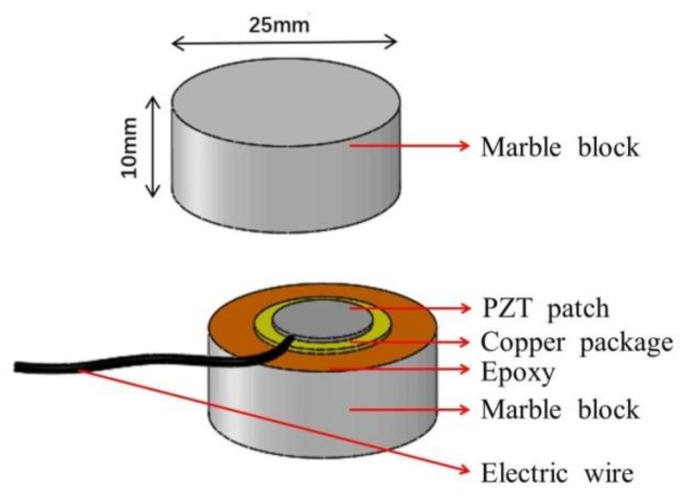
Structure of an SA.

**Figure 3 sensors-20-01376-f003:**
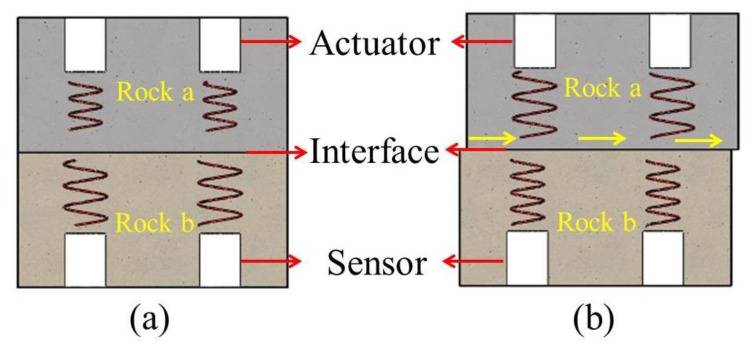
Schematic of active sensing method based on piezoelectric SA: (**a**) before shear failure and (**b**) after shear failure.

**Figure 4 sensors-20-01376-f004:**
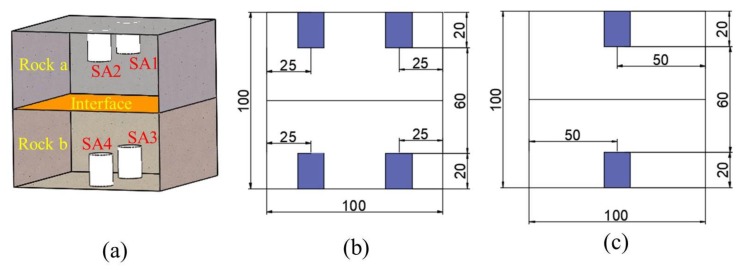
Three-dimensional schematic of specimen and specific size drawing: (**a**) Three-dimensional diagram and (**b**) Front-view and (**c**) Side-view drawings (units: mm).

**Figure 5 sensors-20-01376-f005:**
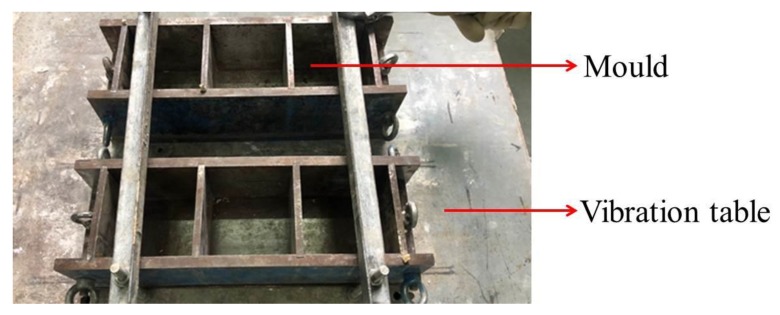
Mould and vibration table.

**Figure 6 sensors-20-01376-f006:**
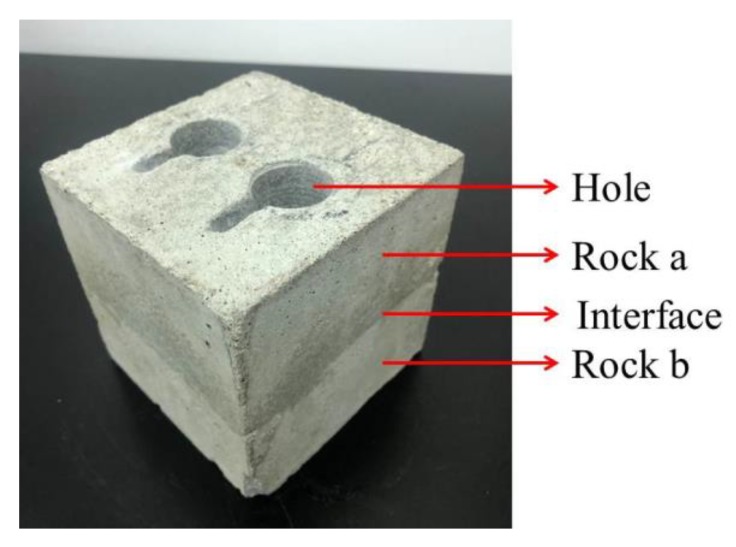
Schematic of borehole specimen.

**Figure 7 sensors-20-01376-f007:**
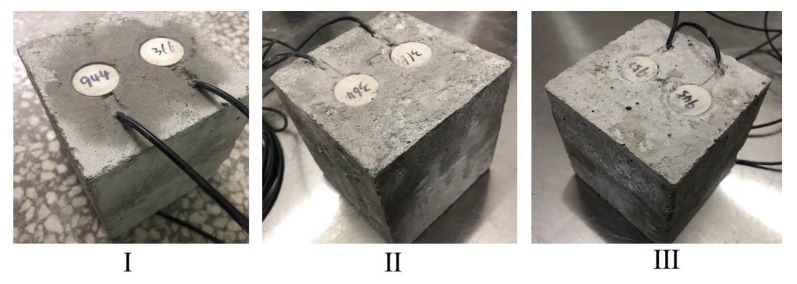
Composite specimens after installation of SAs.

**Figure 8 sensors-20-01376-f008:**
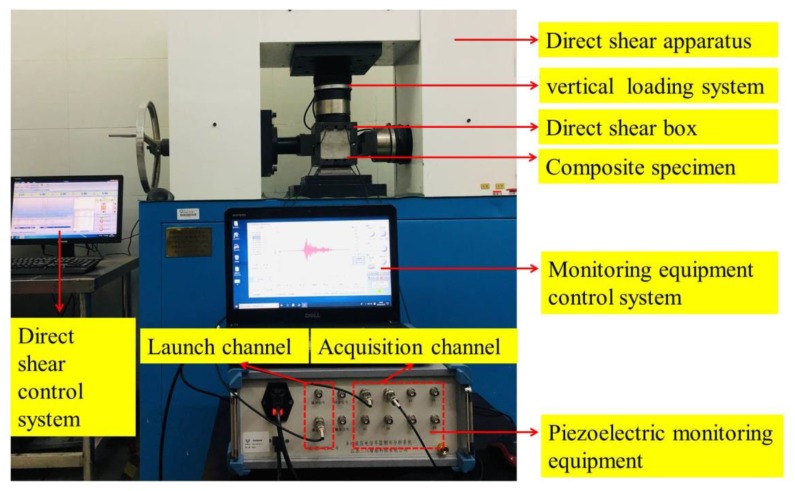
Overall test system.

**Figure 9 sensors-20-01376-f009:**
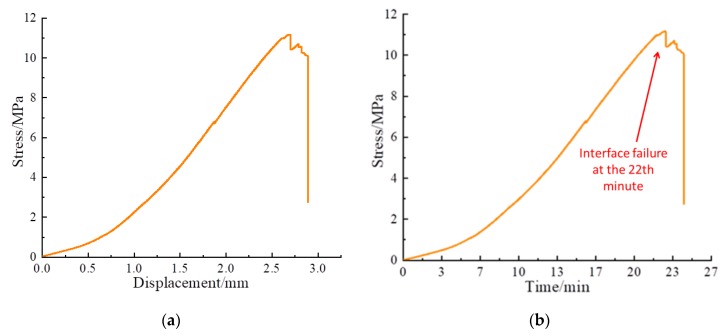
Shear test curve of Composite specimen I: (**a**) Shear stress-displacement curve; (**b**) Shear stress-time curve.

**Figure 10 sensors-20-01376-f010:**
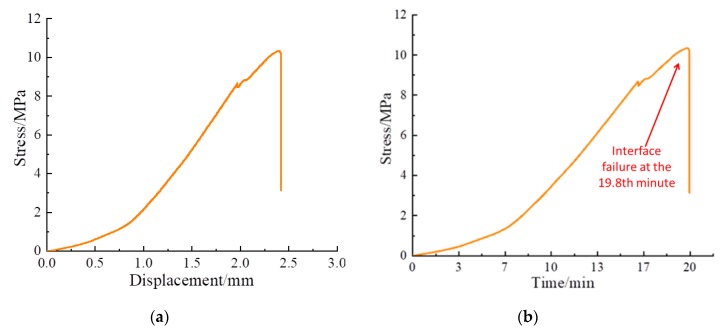
Shear test curve of Composite specimen Ⅱ: (**a**) Shear stress-displacement curve; (**b**) Shear stress-time curve.

**Figure 11 sensors-20-01376-f011:**
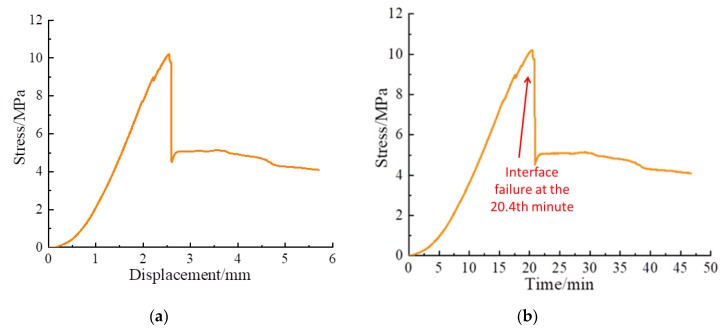
Shear test curve of Composite specimen Ⅲ: (**a**) Shear stress-displacement curve; (**b**) Shear stress-time curve.

**Figure 12 sensors-20-01376-f012:**
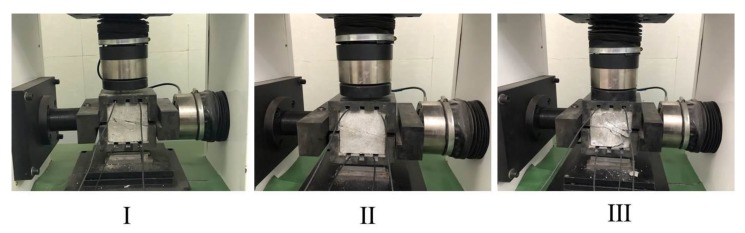
Composite specimens after shear failure.

**Figure 13 sensors-20-01376-f013:**
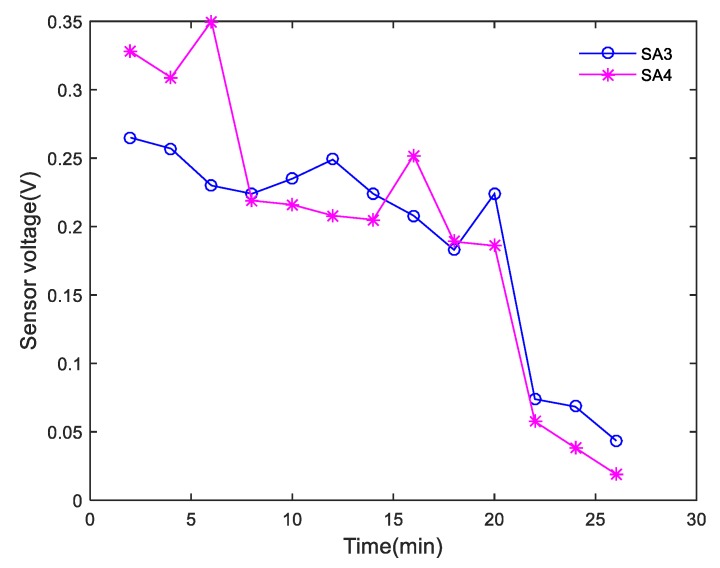
Peak curve of the amplitude of each received SA signal for Composite specimen I.

**Figure 14 sensors-20-01376-f014:**
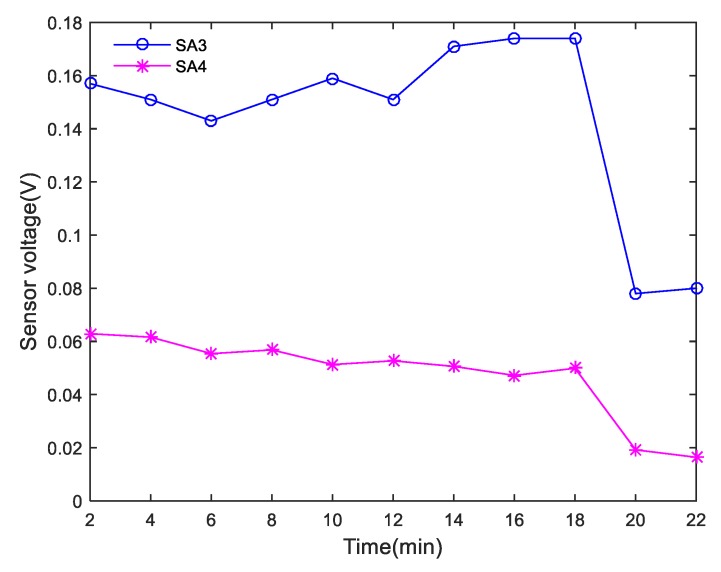
Peak curve of the amplitude of each received SA signal for Composite specimen II.

**Figure 15 sensors-20-01376-f015:**
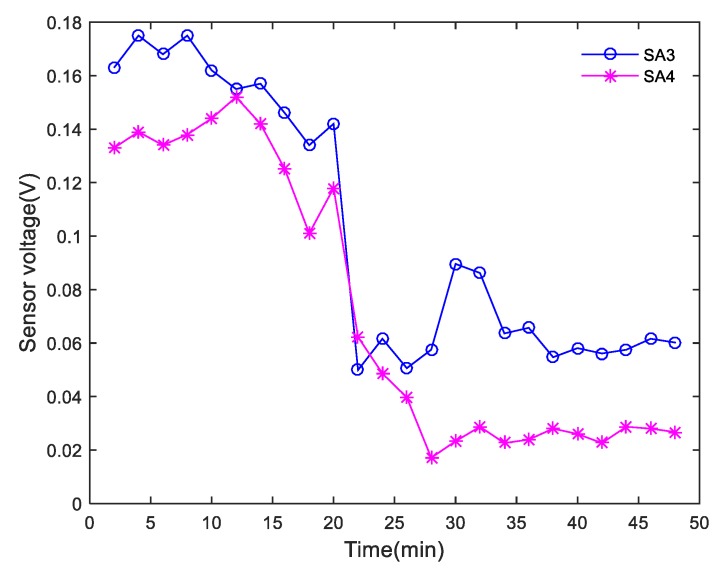
Peak curve of amplitude of each received SA signal for Composite specimen III.

**Figure 16 sensors-20-01376-f016:**
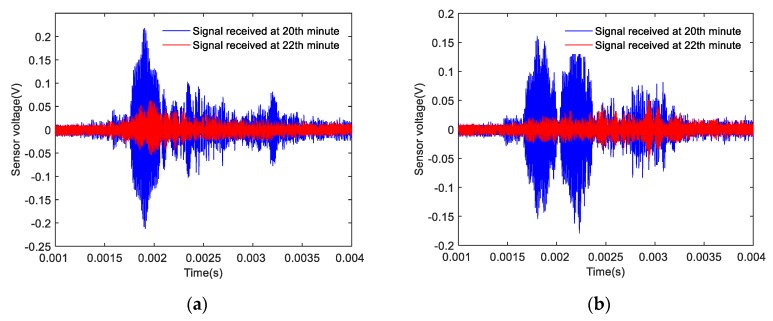
Time-domain signal of Composite specimen I before and after damage: (**a**) Measured signal by SA 3; (**b**) Measured signal by SA 4.

**Figure 17 sensors-20-01376-f017:**
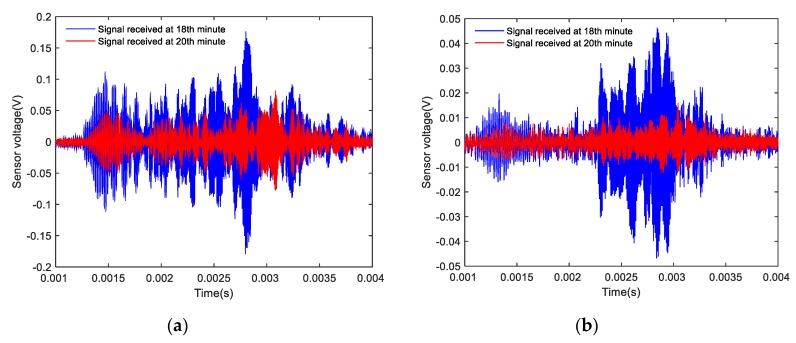
Time-domain signal diagram of Composite specimen II before and after damage: (**a**) Measured signal by SA 3; (**b**) Measured signal by SA 4.

**Figure 18 sensors-20-01376-f018:**
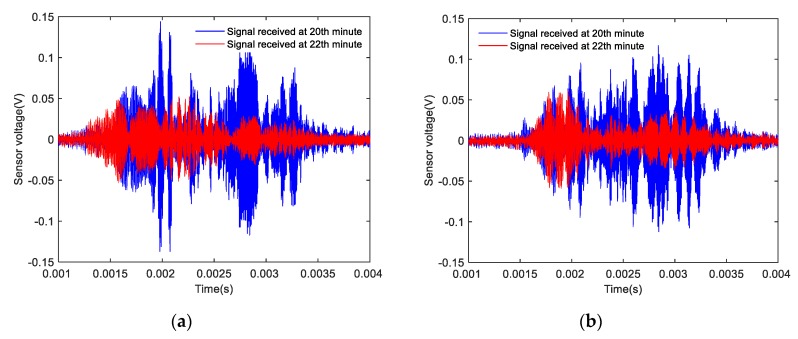
Time-domain signal diagram of Composite specimen III before and after damage: (**a**) Measured signal by SA 3; (**b**) Measured signal by SA 4.

**Figure 19 sensors-20-01376-f019:**
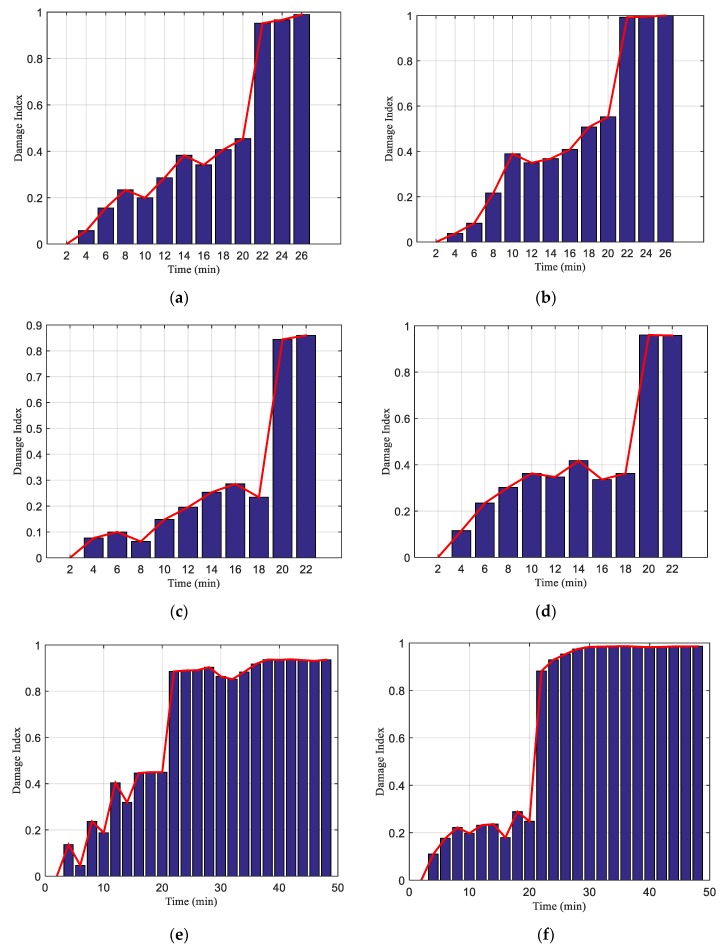
Shear failure damage index of Composite specimen monitored by SAs: (**a**) SA 3 in Composite specimen I; (**b**) SA 4 in Composite specimen I; (**c**) SA 3 in Composite specimen II; (**d**) SA 4 in Composite specimen II; (**e**) SA 3 in Composite specimen III; (**f**) SA 4 in Composite specimen III.

**Table 1 sensors-20-01376-t001:** Material ratio and mechanical parameters.

No.	Elastic Modulus	Cohesive Forces/MPa	The Angle of Internal Friction/*θ*	Tensile Strength /MPa	Plaster-Gray Ratio	Sand-Rubber Ratio	Water-Cement Ratio
a	3.62	4.36	45.65	1.80	0.30	0.65	0.39
b	4.58	5.40	49.85	2.79	0.15	0.50	0.36

**Table 2 sensors-20-01376-t002:** Damage index of composite specimen at failure.

The Number of Composite Specimen	Ⅰ	Ⅱ	Ⅲ
The time of Monitoring (min)	22	20	22
Damage index of SA 3	0.9520	0.8445	0.8857
Damage index of SA 4	0.9922	0.9605	0.8818

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
