# Peer review of "Experimental Research on Shear Failure Monitoring of Composite Rocks Using Piezoelectric Active Sensing Approach"

_sensors, 2020, doi:10.3390/s20051376_

Round 1

Reviewer 1 Report

Dear Authors:

In this manuscript, the authors have studied the shear failure monitoring of composite rocks throughout piezoelectric active sensing method. The main part in this study is piezoelectric smart aggregates (SAs) made from PZT, one typical piezoelectric ceramic. Although this work is practical and interested, the result analysis is a little rough. Firstly, as shown in schematic illustration in Figure 4, two SAs are symmetrically placed at the top and bottom part, e.g. SA1 @ SA2 on the top and SA3@ SA4 on the bottom. However, all shear failure results are obtained from the SA3 and SA4 on the bottom. At the moment, how about the results of SA1 and SA2? Another puzzle is whether the weight of top part affect the sensing system (SA3 and SA4), especially cracking behaviors on three testing samples, as shown in Figure 12. To sum, my suggestion is major revision. Meanwhile, there are some other questions as followed.

  1. According to Figure 9-11, the interface failures of three samples happen at different time, e.g. 22, 19.8, and 20.4 minute, although all samples are made from the similar materials. Please explain this in text.
  2. In Figure 12, three composite specimens exhibit different crack behavior after shear failure testing, especially the crack position. Given all composites are the same, what is the reason?
  3. From Figure 13-15, both sample I and III show the matched curve based on SA 3 and SA 4. But, it is distinct in sample II. There is a big gap between the SA 3 and SA 4. Please explain this.
  4. Figure 19-24 display the shear failure damage indices of all three specimens at SA3 and SA4. At the moment, how about the results of SA1 and SA2. Firstly, at the top and bottom position, whether does the weight of top part affect the testing results of SA3 and SA4? Secondly, given the depths of the holes for SAs are 26mm and 22 mm, while the total width of SA is 20 mm, whether does it play a role on the final signals?

Reviewer 2 Report

This paper is interesting and relevant. The conclusions are consistent. I have minor comments:

1. The materials and methods can be better described. For example, the authors could add the properties, such as density, elasticity modulus, piezoelectric strain among others, of the PZT patches used in this study.
What are the characteristics of the electric wire used in the SA Structure? Also are necessary more information on the connector type used.

This type of information is important for the experiment can be repeated.

2. The subsection "Test device and parameter setting" can be improved. For example, what is the manufacturer and model of the piezoelectric monitoring equipment used? The overall test system is shown in Fig.8. However, the description of the experimental procedure is poor. Lack information on how the measurements were performed and on how the data were collected.

3. The quality and resolution of the Figures 9, 10, 11, 16, 17 and 18 can be improved. Also the font size used in the axes of these figures must be increased. In figures from 16 to 18, the description text used to difference the blue and red signals is unreadable.

4. I suggest to organize the Figures from 19 to 24 in only one figure (a, b, c, d, e and f). For comparison of the results, it is better to organizate these figures in two columns and three lines.

5. The discussion of results shown in subsections 4.2 and 4.3 can be improved. A suggestion is the authors will compare their results with those reported in literature.

Round 2

Reviewer 1 Report

Dear Authors:

The authors carefully have revised the corresponding parts in text, according to my comments. Now it looks better. So it can be accepted in current formation.

Author Response

Thank you very much. We have revised it again.